# Recognition of a Novel Gene Signature for Human Glioblastoma

**DOI:** 10.3390/ijms23084157

**Published:** 2022-04-09

**Authors:** Chih-Hao Lu, Sung-Tai Wei, Jia-Jun Liu, Yu-Jen Chang, Yu-Feng Lin, Chin-Sheng Yu, Sunny Li-Yun Chang

**Affiliations:** 1The Ph.D. Program of Biotechnology and Biomedical Industry, China Medical University, Taichung 404333, Taiwan; chlu@mail.cmu.edu.tw (C.-H.L.); u107305606@cmu.edu.tw (J.-J.L.); u107311201@cmu.edu.tw (Y.-J.C.); 2Department of Medical Laboratory Science and Biotechnology, China Medical University, Taichung 404333, Taiwan; 3Graduate Institute of Biomedical Sciences, China Medical University, Taichung 404333, Taiwan; 4Department of Neurosurgery, China Medical University Hospital, Taichung 404332, Taiwan; zaireeriaz@gmail.com; 5Department of Medical Laboratory Science and Biotechnology, Asia University, Taichung 413305, Taiwan; yflin@asia.edu.tw; 6Department of Information Engineering and Computer Science, Feng Chia University, Taichung 407102, Taiwan; yucs@fcu.edu.tw

**Keywords:** glioblastoma, astrocytoma, oligodendrocytoma, glioma, biomarker signature, gene signature, survival, machine learning

## Abstract

Glioblastoma (GBM) is one of the most common malignant and incurable brain tumors. The identification of a gene signature for GBM may be helpful for its diagnosis, treatment, prediction of prognosis and even the development of treatments. In this study, we used the GSE108474 database to perform GSEA and machine learning analysis, and identified a 33-gene signature of GBM by examining astrocytoma or non-GBM glioma differential gene expression. The 33 identified signature genes included the overexpressed genes *COL6A2*, *ABCC3*, *COL8A1*, *FAM20A*, *ADM*, *CTHRC1*, *PDPN*, *IBSP*, *MIR210HG*, *GPX8*, *MYL9* and *PDLIM4*, as well as the underexpressed genes *CHST9*, *CSDC2*, *ENHO*, *FERMT1*, *IGFN1*, *LINC00836*, *MGAT4C*, *SHANK2* and *VIPR2*. Protein functional analysis by CELLO2GO implied that these signature genes might be involved in regulating various aspects of biological function, including anatomical structure development, cell proliferation and adhesion, signaling transduction and many of the genes were annotated in response to stress. Of these 33 signature genes, 23 have previously been reported to be functionally correlated with GBM; the roles of the remaining 10 genes in glioma development remain unknown. Our results were the first to reveal that GBM exhibited the overexpressed *GPX8* gene and underexpressed signature genes including *CHST9*, *CSDC2*, *ENHO*, *FERMT1*, *IGFN1*, *LINC00836*, *MGAT4C* and *SHANK2*, which might play crucial roles in the tumorigenesis of different gliomas.

## 1. Introduction

Brain tumors are among the most feared and deadliest of all forms of cancers. Primary brain tumors are categorized as glial (arising from glial cells) or nonglial (derived from diverse brain structures including nerves, blood vessels and glands), benign or malignant. Gliomas produced by glial cells are the most common and prevalent type of central nervous system (CNS) tumors. According to their histological features, gliomas are classified as astrocytomas (derived from astrocytes), oligodendroglial tumors, or ependymomas and are assigned World Health Organization (WHO) grades I–IV, according to the presence of anaplastic features indicating varying degrees of malignancy [1]. The most common astrocytomas are pilocytic astrocytomas (WHO grade I), diffuse astrocytomas (grade II), anaplastic astrocytomas (grade III) and glioblastoma multiforme (GBM, grade IV).

In the most recent US population-based cancer registry data on primary brain and other CNS tumors diagnosed between 2014 and 2018, GBM was the most common (14.3% of all tumors and 49.1% of malignant tumors) [2]. GBM is a very aggressive tumor, with a rapid growth rate, as well as high biological and genetic heterogeneity. The prognosis is very poor, with a median survival of only 8 months [2]. Overall 1-, 2- and 5-year relative survival rates for GBM were 40.9%, 6.6% and 4.3%, respectively, amongst 86,355 cases of primary malignant and nonmalignant CNS tumors diagnosed in the USA between 2014 and 2018 [2]. Current standard treatment for GBM consists of maximal surgical resection followed by aggressive chemoradiotherapy using temozolomide, but this fails to prevent the local recurrence of almost all GBM tumors [3]. It is hoped that ongoing explorations into future therapeutic strategies, such as targeted inhibitors of specific molecular processes, nanotherapy and immunotherapy, will improve GBM treatment [3].

The heterogeneity of GBM tumors complicates their diagnoses, predictions of outcomes and decisions on treatment strategies. Recent investigations using biomedical imaging technologies to explore GBM clinical and histological features reflect the aggressiveness of such tumors and provide valuable information for improving the accuracy of prognosis [4,5,6]. In addition to the histological features, genetic features, such as DNA methylation data and molecular biomarkers, provide insights into the mechanisms underlying GBM and its diagnosis and have been included in the WHO classification of brain tumors [7,8]. Gene- and protein-based signatures that have been identified in bioinformatics studies have revealed gene and protein biomarkers of GBM survival [9,10,11]. Identified molecular biomarkers of GBM include immune-related molecules [10], cytoskeletal proteins [12,13], nonprotein-coding RNAs [14] and signal-related molecules revealing signaling pathways that play crucial roles in GBM, i.e., EGFR [15,16], VEGF [17], sonic hedgehog (SHH) [18,19], Notch [20] and Wnt/β-catenin signaling [21,22]. These molecular biomarkers not only greatly assist with the diagnosis and classification of GBM, but also have enormous potential as drug targets in therapeutic drug development [20,23,24,25,26,27]. However, it is important to realize that while changes in expression levels of many GBM biomarkers might be associated with the suppression of tumor proliferation in GBM, none of these biomarkers are capable of destroying all of the tumor cells [20,23,24,25,26,27]. This suggests that identifying therapeutic strategies involving biomarkers with potential to destroy tumor cells in GBM remains an urgent task.

In recent years, large-scale human genomic data emerging from several projects have been stored in public databases, including the National Center for Biotechnology Information Gene Expression Omnibus (NCBI GEO) repository [28,29], The Cancer Genome Atlas (TCGA) [30] and the Catalogue Of Somatic Mutations In Cancer (COSMIC) database [31]. As some researchers have proposed, the efficient storage and processing of big data may help us to extract essential information and to understand the complicated mechanisms of life [32]. Machine learning methods offer us the capacity to leverage big data and to analyze complex problems, so that we may clarify information conveyed by these data. This study has consulted records from the Rembrandt (REpository for Molecular BRAin Neoplasia DaTa) brain cancer patient-derived dataset, a joint project established by the US National Institutes of Health (NIH) National Cancer Institute (NCI) and National Institute of Neurological Disorders and Stroke (NINDS) [33]. The genomic data from this project are available on the open-access Georgetown Database of Cancer (G-DOC) platform and in the NCBI GEO repository as a super series GSE108474 [33]. Using gene set enrichment analysis (GSEA) and support vector machine (SVM) learning analysis, our analysis of the biological signatures identified a novel 19-gene GBM signature in the glioblastoma and astrocytoma gene expression data (GBM1 dataset) and a novel 17-gene GBM signature in the glioblastoma and nonglioblastoma patient samples (GBM2 dataset).

Our findings not only provide a new point of reference for the prognostic prediction of GBM, but also contributed to the understanding of molecular mechanisms in GBM development. Furthermore, these novel signature genes might potentially be exploited as therapeutic targets for GBM.

## 2. Results

### 2.1. Recognition of Novel Gene Signatures in GBM

This study employed the machine learning method to construct prediction models and select critical genes from two datasets, GBM1 and GBM2. Each prediction model was subjected to the five-fold cross-validation technique to evaluate predictive performances. Table 1 presents predictive performances from the GBM1 and GBM2 datasets, optimized with the Matthews correlation coefficient (MCC). The performance of the GBM1 dataset was slightly superior to that of the GBM2 dataset, with an accuracy of 92%, an MCC of 0.83 and an F1 score of 0.93, which might be because the GBM1 dataset consisted of only two tumor types (glioblastoma multiforme and astrocytoma), whereas the GBM2 dataset had five (glioblastoma multiforme, astrocytoma, oligodendroglioma, mixed and unclassified).

Gene scores ranged from 0 to 3, where the highest value had the most impact, representing the most frequently selected gene. A total of 19 and 17 genes scored ≥2 with their gene signatures in the GBM1 and GBM2 datasets, respectively (Table 2). Three genes, including *IGFN1*, *MGAT4C* and *ADM*, were selected in both datasets. Thus, a total of 33 GBM gene signatures were identified. Twelve of these were overexpressed in GBM, including the *MIR210HG* nonprotein-coding gene and 11 protein-coding genes (*COL6A2*, *ABCC3*, *COL8A1*, *FAM20A*, *ADM*, *CTHRC1*, *PDPN*, *IBSP*, *GPX8*, *MYL9* and *PDLIM4*) (Table 2). The underexpressed GBM genes included the *LINC00836* nonprotein-coding gene and 20 protein-coding genes (*FERMT1*, *DLL3*, *P2RY12*, *CHST9*, *IFGN1*, *CSDC2*, *ETNPPL*, *VIPR2*, *MGAT4C*, *DLL1*, *TNR*, *GDF10*, *IRX2*, *SHANK2*, *ENHO*, *LUZP2*, *DPP10*, *CDHR1*, *AKR1C3* and *SCG3*) (Table 2).

The microarray expression data of the 19 GBM1 and 17 GBM2 selected genes are shown as heatmaps in Figure 1a, b, respectively. The GSEA analysis identified intersections of 42 overexpressed genes and 35 underexpressed genes from the combined GBM1 and GBM2 datasets (Figure 2). In the 33-gene signature, our methods selected only one overexpressed (*ADM*) and two underexpressed (*IGFN1* and *MGAT4C*) intersection genes. Twelve overexpressed and 21 underexpressed signature genes were recognized from 58 overexpressed and 65 underexpressed genes in both GBM1 and GBM2 datasets. Of particular interest was that only three intersection genes were recognized as signature genes from 77 intersection genes, while 30 genes were recognized by 46 genes that were either in the GBM1 or GBM2 dataset.

### 2.2. Functional Analysis

We performed the protein functional analysis with CELLO2GO [34], a web server that uses BLAST homology searching approaches to discern functional gene-ontology (GO) annotation. Two GO functional annotation categories, biological processes (BPs) and molecular functions (MFs) were retrieved for each selected protein-coding gene signature. An analysis of the selected 31 protein-coding gene signatures revealed that they were annotated to 50 and 24 GO-enriched groups in the BP and MF categories, respectively (Appendix A). Statistically, 15 GO terms for the BP category retrieved over 10 annotated proteins. The top five GO terms retrieved were “anatomical structure development”, “response to stress”, “cell differentiation”, “signaling transduction” and “cell adhesion” (Figure 3a; Appendix A). Ten of the retrieved GO terms in the MF category had more than five annotated proteins; the top three were “ion binding”, “protein binding” and “structural molecule activity” (Figure 3b; Appendix A). Notably, more than half of the proteins encoded by signature genes were annotated to the “ion binding” and “protein binding” GO terms associated with the MF category.

The KEGG database [35] was used to identify potential signaling pathways involving the selected protein-coding gene signatures. Four of the retrieved signaling pathways (“focal adhesion”, “PI3K-Akt signaling pathway”, “ECM-receptor interaction” and “human papillomavirus infection”) each had more than two annotated proteins (Figure 4a; Appendix A). CELLO2GO also predicted protein subcellular localization. Proteins encoded by the 31 gene signatures were mostly located in the nuclear (18 proteins), extracellular (8 proteins) or cytoplasmic regions (8 proteins), or were sited within the plasma membrane (7 proteins) (Figure 4b; Appendix A). CELLO2GO predicted the locations of proteins SCG3 and CHST9 in the endoplasmic reticulum and mitochondria, respectively.

### 2.3. Roles of the Overexpressed GBM Gene Signatures

A literature search revealed that all of the selected overexpressed gene signatures were differentially expressed in gliomas and associated with either poor prognosis or low survival rates, except for the *GPX8* gene (Table 3). Of the 12 overexpressed genes, *ABCC3*, *COL8A1*, *IBSP* and *PDPN* are reportedly associated with GBM [10], while *CTHRC1*, *PDLIM4* and *MYL9* are associated with high-grade and malignant gliomas, respectively [36,37]. These findings were consistent with the evidence of their overexpression in GBM revealed by our study, except for *ABCC3*, which in one study was apparently expressed at lower levels in GBM tissue compared with normal brain tissue samples [38]. Correlations between the *ABCC3*, *COL6A2* and *FAM20A* genes and low-grade glioma suggested that these genes might play more complex roles in gliomas [38,39,40].

### 2.4. Roles of the Underexpressed GBM Gene Signatures

Of the 21 underexpressed genes, 13 (*AKR1C3*, *CDHR1*, *DLL1*, *DLL3*, *DPP10*, *ETNPPL*, *GDF10*, *IRX2*, *LUZP2*, *P2RY12*, *SCG3*, *TNR* and *VIPR2*) are reportedly associated with gliomas (Table 4). These genes are underexpressed in GBM and overexpressed in low-grade gliomas, including diffuse astrocytomas, anaplastic astrocytoma and oligodendrocytoma. Observed correlations between decreasing levels of expression of these genes and increasingly malignant gliomas suggested that these genes might be worth targeting in glioma treatment [20,24,25,26,27,47,48,49]. No published evidence was available as to the roles of the remaining eight underexpressed gene signatures (*CHST9*, *CSDC2*, *ENHO*, *FERMT1*, *IGFN1*, *LINC00836*, *MGAT4C* and *SHANK2*) regarding glioma pathogenesis or prognosis.

### 2.5. Protein–Protein Interaction Network Analysis Using STRING

We used STRING (Search Tool for the Retrieval of Interacting Genes; https://string-db.org, accessed on 1 March 2022) to assess potential interactions of the proteins encoded by the 31 GBM gene signatures. The predicted protein–protein interaction (PPI) networks contained four linkage groups, modules I, II, III and IV, containing 3, 4, 2, and 15 nodes, respectively (Figure 5). All three members of module I (IRX2, DLL1 and DLL3) were encoded by underexpressed genes in GBM and all were annotated with the GO terms of “embryo development”, “anatomical structure formation involved in morphogenesis”, “anatomical structure development”, “cell differentiation” and “reproduction” in the BP category (Appendix A).

Members in module II (TNR, SCG3, ADM and VIPR2) were all annotated to the GO terms of “transport”, “vesicle-mediated transport” and “response to stress” in the BP category. Module III, containing a two-node linkage of AKRC3 and ABCC3, was also annotated to the GO term “response to stress”, in addition to the “lipid metabolic”, “biosynthetic process”, “catabolic process” and “metabolic process” GO terms in the BP category (Appendix A).

Of particular interest was that the largest module (IV) of 15 proteins was centrally linked by COL6A2, COL8A1 and CTHRC1, a triangular interactive network of collagen-related proteins that were all overexpressed in GBM and have previously been reported as correlating with increasing tumor cell invasion, migration and adhesion in GBM (Table 3, Figure 4) [10,39,42]. The other five nodes that reportedly correlate with GBM malignancy (PDLIM4, MYL9, PDPN, IBSP and GPX8) [10,13,36,37,46] were all (with the exception of GPX8) overexpressed in the GBM gene signatures (Figure 4). Among the 15 node proteins, only GPX8 and DPP10 were not annotated to the GO term “anatomical structure development”. The majority of nodes (7 of 10) in the COL6A2 and COL8A1 side chains were annotated to the GO term “cell adhesion” in the BP category (MYL9, GDP10 and P2RY12 were not).

## 3. Discussion

This study found a novel 33-gene GBM signature by using the GSE108474 database to compare differentially expressed genes amongst GBM and astrocytomas or non-GBM gliomas. Notably, there were more overexpressed than underexpressed intersection genes in the selected genes of both the GBM1 and GBM2 datasets. This was not unexpected because these datasets shared the same GBM patient samples and there might be a greater association between the overexpressed genes and GBM, while the underexpressed genes could be more associated with the partially overlapping gliomas between the datasets. Our results implied that our methodological approaches were independent and were not biased by the intersection genes.

Of the 11 protein-coding, overexpressed signature genes, only *GPX8* was not associated with either GBM or glioma (Table 3). GPX8 is an endoplasmic reticulum-resident protein that plays an important role in various cancers, such as gastric, breast and non-small cell lung cancer [55,56,57]. FoxC1-induced transcriptional activation of GPX8 activates Wnt signaling and subsequent gastric cancer cell proliferation [55]. The Wnt/β-catenin signaling pathway has pleiotropic functions in neurogenesis and is one of the main signaling pathways in glioma tumorigenesis [58,59]. High levels of FoxC1 expression have been found in gliomas and FoxC1 may regulate epithelial-to-mesenchymal transition (EMT) via Wnt/β-catenin signaling [60]. Furthermore, evidence has shown that GPX8 maintains an aggressive breast cancer phenotype by regulating the interleukin 6 (IL-6)/JAK/STAT3 signaling pathway [57], which is hyperactivated in many different malignancies and is generally linked to a poor prognosis [61]. Indeed, aberrantly activated STAT3 signaling is positively correlated with tumor grade and survival rates of patients with GBM [62]. Future research could usefully investigate whether the overexpression of *GPX8* in GBM is activated by FoxC1 and consequently regulates GBM tumorigenesis via the Wnt/β-catenin pathway or via GPX8/IL-6/STAT3 signaling.

The identification of the underexpressing genes in GBM might contribute to the development of a therapeutic strategy. Twelve of the 13 protein-coding, underexpressed signature genes were associated with GBM or other gliomas (Table 4). Many of them, such as *CDHR1*, *DLL1*, *DLL3* and *SCG3*, have been reported to be potential therapeutic targets for glioma treatment [20,23,24,25,26,27]. We failed to find any reports in the literature revealing the role of the other seven selected protein-coding, underexpressed signature genes (*CHST9*, *CSDC2*, *ENHO*, *FERMT1*, *IGFN1*, *MGAT4C* and *SHANK2*). However, all reportedly play crucial roles in various tumors and thus may also play important roles in gliomas.

Genetic variants of *CHST9* contribute to the prognosis of triple-negative breast cancer [63] and the copy number variants of *CHST9* are associated with hematologic malignancies [64]. *CSDC2* has been shown to be a potential diagnostic biomarker for early-onset colorectal cancer [65] and prostate cancer [66]. The protein encoded by the *ENHO* gene, adropin, is a secreted peptide hormone related to energy homeostasis [67] and is reportedly associated with the pathogenesis of endometrium cancer [68]. Evidence has suggested that *FERMT1* silencing inhibits oral squamous cell cancer EMT and invasion by inactivating the PI3K/AKT signaling pathway, one of the most important pathways in GBM [69]. As for *IGFN1*, although the biological function of this gene remains unclear, *IGFN1* expression has been associated with susceptibility to primary retroperitoneal liposarcoma and renal cell carcinoma, and the radiotherapy response in non-small cell lung cancer [70,71,72]. *MGAT4C* overexpression in benign and cancer prostate cell lines significantly increases their proliferation and migration, and increasingly higher *MGAT4C* transcript levels are associated with prostate cancer progression [73].

*SHANK2* is the most frequently amplified gene on 11q13, a major tumor amplicon in human cancer [74]. *SHANK2* plays an evolutionarily conserved role in the regulation of Hippo signaling [74], which promotes tumorigenesis and metastasis of several cancers including GBM, although scant data exist on the role of Hippo signaling in brain tumors [75,76]. *SHANK2* may be important for the development of GBM [74]. *VIPR2*, also known as VIP and PACAP receptor 2, is a receptor subtype for pituitary adenylate cyclase-activating polypeptide (PACAP) [77]. VIP and PACAP are neurotransmitters and neuromodulators that regulate neurons in various aspects, such as neuronal division, differentiation and survival [78]. Under normal culture conditions, VIP and PACAP induce the proliferation of rat GBM-derived C6 glioma cells, while under serum-starved conditions, VIP and PACAP possess antiproliferative properties [78]. As the receptor of VIP and PACAP, VIPR2 may regulate GBM development [77]. Importantly, the associations of these eight underexpressed GBM genes with various tumors imply that they may also play crucial roles in glioma development.

This study identified two nonprotein-coding RNA genes, *MIR210HG* and *LINC00836*, as GBM signature genes. *MIR210HG* was overexpressed in GBM compared with non-GBM gliomas. Previous research has found *MIR210HG* within a nine-EMT-related lncRNA signature in patients with glioma, while other evidence suggests that *MIR210HG* is an important diagnostic biomarker for glioma [44,45]. No reports have documented the underexpression of *LINC00836* in GBM compared with astrocytomas. Nevertheless, although no evidence exists as to the differential expression of *LINC00836* in gliomas, this gene is known to be a key Alzheimer’s disease-related immune hub gene that potentially contributes to immune-related phenomena in Alzheimer’s disease by regulating other immune-related hub genes in the Alzheimer’s brain [79]. Further analysis should explore the role of *LINC00836* in glioma pathogenesis.

## 4. Materials and Methods

### 4.1. Brain Tumor Dataset

GEO Series Experiment 108474 (GSE108474), which was part of the Rembrandt brain cancer project from 2004 to 2006, contains clinical and biospecimen data from 671 patients collected from 14 institutions. Records for 550 patients that had both gene expression and clinical metadata are held in the NCBI GEO database [33]. Table 5 lists these 550 samples by tumor types obtained from the NCBI GEO repository. Two datasets were collected based on the types of brain tumors documented in the metadata. The first dataset (GBM1) contained 148 astrocytoma and 228 GBM samples, while the second dataset (GBM2) contained 228 GBM samples and 227 other brain carcinoma samples, including 148 astrocytomas, 11 mixed, 67 oligodendrogliomas and 1 unclassified tumor.

### 4.2. Gene Feature Generation

The gene expression data of 550 patient samples in GSE108474 were sequenced on the GEO platform 570 (GPL570, Affymetrix Human Genome U133 Plus 2.0 Array [HG-U133_Plus_2]) containing 54,675 human probsets. The gene expression values from the microarray data were first normalized using MAS5 [80] (linear scaling) before adopting the log base 2 scale. The GSEA method [81] was then used for the preliminary screening of genes, to determine whether any statistically significant differences were apparent between GBM and other types of brain tumors. The normalized microarray data from each sample were first labeled by categorical classes, before comparing the genes inside the microarray with the *a priori* defined gene set from the REACTOME [82] subset of canonical pathways curated by the Molecular Signatures Database (MSigDB) [83]. We used the default algorithm signal-to-noise ratio to compare differences in gene expression amongst different phenotypes and ranked the genes based on the value of the formula. To evaluate the distribution of genes in the REACTOME gene set across the entire ranked list, the GSEA method was used to perform a running sum statistic progressing down the ranked list: if a gene in the ranked list belonged to the gene in the REACTOME set, the GSEA increased the accumulative score and reduced it when genes did not belong. In this process, the maximum value was denoted as the enrichment (EC) score. The EC score reflected which genes in the REACTOME set were overexpressed at the top or bottom of the ranked list. Finally, we obtained the top 50 over- or underexpressed genes ranked by the EC score as our classification features for input into the machine learning method.

### 4.3. Selection of Critical Genes

The genetic algorithm (GA) [84,85,86] was used to select critical genes and optimize classification performance. The GA procedures in this work were as follows: in the initial population, we randomly generated 80 solutions ($S_{i},\;i=1,\ldots ,80$), where each solution $S_{i}$ was represented as a set of 100-dimensional feature vectors $(f_{j}^{i},\;j=1,\ldots ,100$), indicating the binary representations of 100 genes selected from GSEA. If $f_{j}^{i}=1$, the $j^{th}$ gene was kept; if $f_{j}^{i}=0$, the gene $j^{th}$ was eliminated for feeding into the SVM, a supervised learning method that used the principle of statistical risk minimization to estimate the hyperplane of a classification. All SVM calculations were performed using LIBSVM (version 3.24) [87,88], with the radial basis function (RBF) kernel. The parameters (penalty and gamma values of the RBF kernel) were both trained by exponentially increasing the grid search from 2^−15^ to 2^15^, incorporating the best values of informative measures with a five-fold cross-validation during model training.

In this work, 200 generations were iterated. For each generation, τ, the three basic mechanisms driving the evolutionary processes were performed consisting of the selection, mutation and crossover processes. The selection operators were defined as $\alpha^{\tau }=\max\left\{S_{1}^{\tau },\ldots ,S_{40}^{\tau },\alpha^{\tau -1}\right\}$ and $\beta^{\tau }=\max\left\{S_{41}^{\tau },\ldots ,S_{80}^{\tau },\beta^{\tau -1}\right\}$. The solutions $\alpha^{\tau }$ and $\beta^{\tau }$ had the best fitness values in each half of 80 solutions and $\alpha^{\tau -1}$ and $\beta^{\tau -1}$ in the previous generation, respectively. The best solutions $\omega^{\tau }$ had the best fitness values between $\alpha^{\tau }$ and $\beta^{\tau }$. Note that for the special case of τ=0, the fitness values of $\alpha^{0}$ and $\beta^{0}$ were defined as 0. A new solution in the next generation τ+1, $S_{i}^{\tau +1}$, was equal to $\alpha^{\tau }$ if i was odd, while $S_{i}^{\tau +1}$ was equal to $\beta^{\tau }$ if i was even.

Four informative measures (Equations (1)–(4)) calculated using five-fold cross-validation during model training were used as the fitness functions in the selection process. They consisted of accuracy (*Acc*), the *MCC*, the F1 score (*F*1), summation of sensitivity and weighted specificity (hybrid), calculated as follows:(1)$$Acc=\frac{TP+TN}{TP+TN+FP+FN}$$
(2)$$MCC=\frac{TP\times TN-FP\times FN}{\sqrt{\left(TP+FP\right)\left(TP+FN\right)\left(TN+FP\right)\left(TN+FN\right)}}$$
(3)$$F1=\frac{2\times Precision\times Sensitivity}{Precision+Sensitivity}$$
(4)Hybrid=Sensitivity+δ×Specificity
where $Precision=\frac{TP}{TP+FP}$, $Sensitivity=\frac{TP}{TP+FN}$, $Specificity=\frac{TN}{TN+FP}$, *TP* represents true-positives, *TN* represents true-negatives, *FP* represents false-positives, *FN* represents false-negatives and δ is the ratio of the number of positives to negatives.

After adopting the selection operators, two types of mutation were applied to all solutions ($S_{i}$s). In the first half of the solutions, every b bit of the vectors was subject to mutation: b=~b, if the mutation rate was less than a mutation threshold $\mu_{mu}=0.1$. In the second half of the solutions, we randomly chose a bit from the 100 vectors subject to mutation: b=~b without any mutation thresholds. The one-point crossover operations were carried out between $S_{2k-1}$ and $S_{2k}$, where k=1,…,40 and proceeded as follows: the feature vectors from λ to 100 of $S_{2k-1}$ and $S_{2k}$ were swapped if the crossover rate was less than the crossover threshold $\mu_{cr}=0.5$, where λ was randomly selected from 1 to 100.

The selection procedure of critical genes, performed by the genetic algorithm, was repeated 10 times with each informative measure. Each repeat produced 200 best solutions $\left(\omega^{\tau },\;\tau =1,\ldots ,200\right)$ for every 200 generations. Thus, we generated a total of 2000 solutions. After deleting the redundant solutions, the top 10 were obtained from the remainder and ranked by fitness value. The selective scores ($r_{j},\;j=1,\ldots ,100)$ of 100 genes were calculated from the top 10 solutions, as follows:(5)$$r_{j}=\frac{1}{4}\overset{4}{\underset{p=1}{\sum }}(\frac{1}{10}\overset{10}{\underset{q=1}{\sum }}f_{j}^{q}+\frac{1}{5}\overset{5}{\underset{q=1}{\sum }}f_{j}^{q}+\overset{1}{\underset{q=1}{\sum }}f_{j}^{q})$$
where p is the specific informative measure (Equation p) used as the fitness function, and q represents the top q-th solution.

## 5. Conclusions

GBM is one of the most common malignant and incurable brain tumors. The molecular mechanisms underlying the tumorigenesis of GBM, a biologically heterogeneous tumor, remain unclear. The identification of a gene signature for GBM may be helpful for the diagnosis, treatment, prediction of prognosis, and even new treatments for GBM. The identification of a 33-gene signature of GBM by GSEA and machine learning analysis revealed reported associations with GBM for 24 of these genes; the roles of the remaining nine in glioma development remain unknown. Our results were the first ever to report that the overexpressed *GPX8* gene and the underexpressed GBM *CHST9*, *CSDC2*, *ENHO*, *FERMT1*, *IGFN1*, *LINC00836*, *MGAT4C* and *SHANK2* genes might play crucial roles in the tumorigenesis of different gliomas.

## Figures and Tables

**Figure 1 ijms-23-04157-f001:**
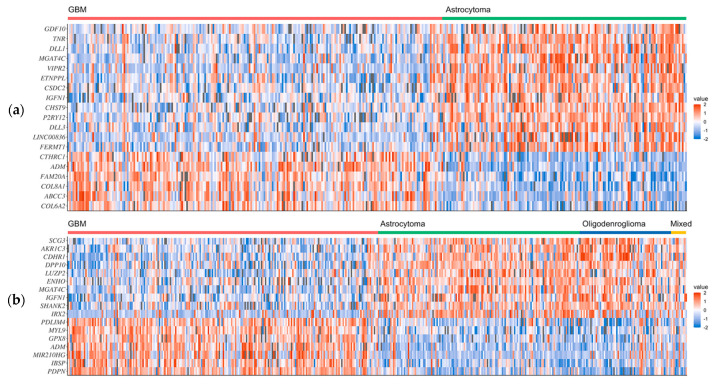
Heatmaps of the selected signature genes of GBM from the (**a**) GBM1 and (**b**) GBM2 datasets.

**Figure 2 ijms-23-04157-f002:**
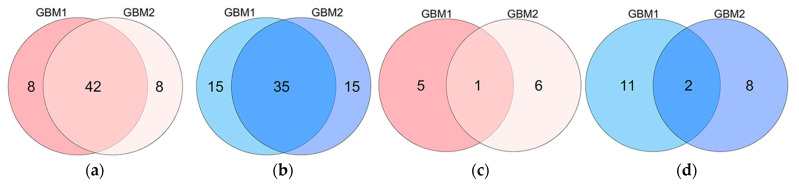
Venn diagrams of the numbers of original (**a**,**b**) and selected (**c**,**d**) genes in the GBM1 and GBM2 datasets, respectively. (**a**,**c**) represents the overexpressed genes in GBM samples; (**b**,**d**) represents the underexpressed genes in GBM samples.

**Figure 3 ijms-23-04157-f003:**
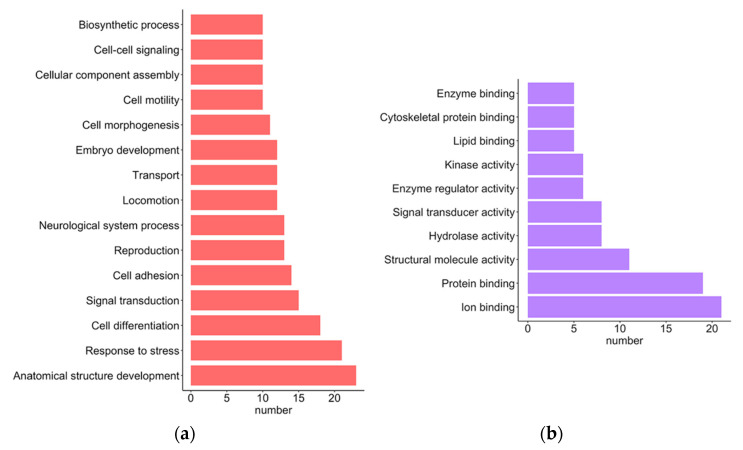
The GO annotations of the proteins encoded by selected signature genes of GBM. The lengths of the bars represent the number of proteins involved. (**a**) The GO terms of “biological process”, which involve more than 10 proteins. (**b**) The GO terms of “molecular function”, which involve more than five proteins.

**Figure 4 ijms-23-04157-f004:**
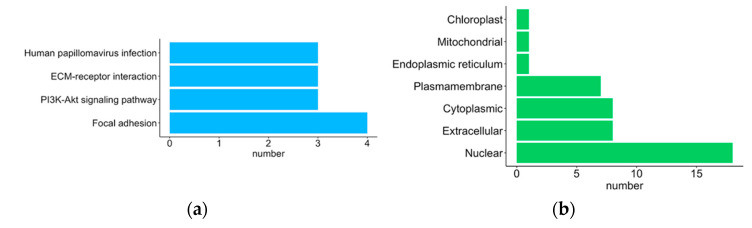
The KEGG pathways and predicted subcellular localization of the proteins encoded by selected signature genes of GBM. The lengths of the bars represent the number of proteins involved. (**a**) The KEGG pathways, which involve more than three proteins. (**b**) Predicted subcellular localization by CELLO2GO.

**Figure 5 ijms-23-04157-f005:**
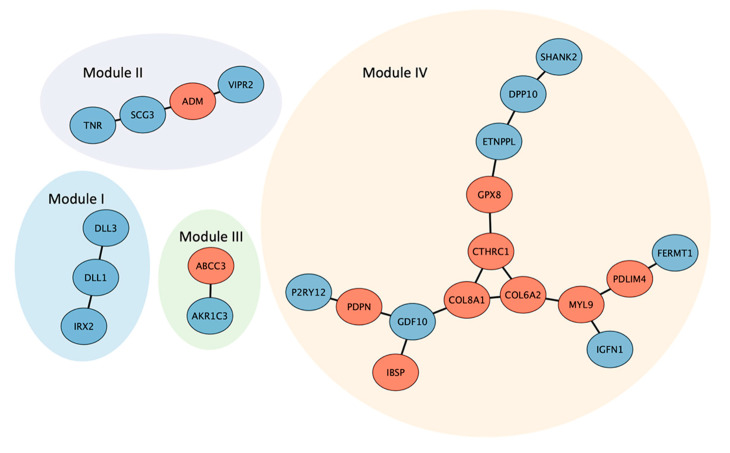
Four linkage modules (I, II, III and IV) of predicted protein–protein interaction networks by STRING. The nodes in red and blue represent the proteins encoded by the over- and underexpressed GBM signature genes, respectively.

**Table 1 ijms-23-04157-t001:** Predictive performances of the GBM1 and GBM2 datasets. All predictions were optimized using the MCC as the fitness function.

Datasets	Accuracy	Sensitivity	Specificity	MCC	Precision	F1 Score
GBM1	0.9176	0.9517	0.8648	0.8265	0.9156	0.9333
GBM2	0.8835	0.8947	0.8722	0.7672	0.8755	0.8850

**Table 2 ijms-23-04157-t002:** Expression signatures of the 33 genes exhibiting high selective scores (≥2) in the GBM1 and GBM2 datasets.

Datasets	Overexpressed Genes	Underexpressed Genes
GBM1	*COL6A2*, *ABCC3*, *COL8A1*, *FAM20A*, *ADM*^1^, *CTHRC1*	*FERMT1*, *LINC00836*^2^, *DLL3*, *P2RY12*, *CHST9, IGFN1*^1^, *CSDC2*, *ETNPPL*, *VIPR2*, *MGAT4C*^1^*, DLL1*, *TNR*, *GDF10*
GBM2	*PDPN*, *IBSP*, *MIR210HG*^2^, *ADM*^1^, *GPX8*, *MYL9*, *PDLIM4*	*IRX2*, *SHANK2*, *IGFN1*^1^, *MGAT4C*^1^, *ENHO, LUZP2*, *DPP10*, *CDHR1*, *AKR1C3*, *SCG3*

^1^ Selected by both the GBM1 and GBM2 datasets. ^2^ Nonprotein-coding RNA genes.

**Table 3 ijms-23-04157-t003:** Roles/functions of overexpressed GBM gene signatures in glioma.

Gene	Encoded Protein	Reported Roles/Functions in Gliomas	References
*ABCC3*	ATP-binding cassette subfamily C member 3	Lower expression in GBM tissue versus normal brain tissue; low expression is associated with low survival rates.	Su et al., 2020 [38]
		Upregulated expression correlates with poor overall survival in GBM.	Jiang et al., 2021 [10]
*ADM*	Proadrenomedullin	Positively regulated by STAT-3 signaling; enhances the migration of astroglioma cells.	Lim et al., 2014 [41]
*COL6A2*	Collagen alpha-2(VI) chain	High expression associated with worse prognosis; induces tumor cell proliferation in recurrent and high-risk low-grade glioma.	Chen et al., 2020 [39]
*COL8A1*	Collagen alpha-1(VIII) chain	High expression correlated with poor overall survival in GBM.	Jiang et al., 2021 [10]
*CTHRC1*	Collagen triple helix repeat containing protein-1	Increased expression in glioma tissue is associated with WHO disease stage; regulates tumor cell invasion, migration and adhesion.	Mei et al., 2017 [42]
*FAM20A*	Psedokinase FAM20A	Biomarker for low-grade glioma; overexpression predicts poor outcomes.	Feng et al., 2021 [40]
		Associated with disrupted immune responses in the GBM microenvironment.	Du et al., 2020 [43]
*GPX8*	Glutathione peroxidase 8	N/A.	
*IBSP*	Integrin-binding bone sialoprotein 2	High expression correlated with poor overall survival in GBM.	Jiang et al., 2021 [10]
*MIR210HG*	Nonprotein-coding gene	Identified as an EMT-related lncRNA in gliomas.	Tao et al., 2021 [44]
		Serves as a biomarker for glioma diagnosis.	Min et al., 2016 [45]
*MYL9*	Myosin regulatory light polypeptide 9	High expression is associated with a poor prognosis and is increased in patients with recurrent disease.	Kruthika et al., 2019 [13]
		The DAPK1-ITPRIP-MYL9 complex promotes the progression of malignant glioma.	Cao et al., 2021 [37]
*PDLIM4*	PDZ and LIM domain protein 4	Biomarker for high-grade glioma.	de Tayrac et al., 2013 [36]
*PDPN*	Podoplanin	Correlated with poor overall survival in GBM.	Jiang et al., 2021 [10]
		Increases tumor cell migration and angiogenesis in malignant glioma.	Grau et al., 2015 [46]

**Table 4 ijms-23-04157-t004:** Roles/functions of the underexpressed GBM gene signatures in glioma.

Gene	Encoded Protein	Reported Roles/Functions in Gliomas	References
*AKR1C3*	Aldo-keto reductase family 1 member C3	A hormone activity regulator and prostaglandin F synthase that is expressed in GBM and oligodendrogliomas; associated with the duration of overall survival in patients with gliomas.	Park et al., 2010 [50]
*CDHR1*	Cadherin-related family member 1	Downregulated in GBM and other gliomas (compared with normal brain tissue); lower expression of CDHR1 is associated with worse clinical prognosis in GBM.	Wang et al., 2021 [24]
*CHST9*	Carbohydrate sulfotransferase 9	N/A.	
*CSDC2*	Cold shock domain-containing protein C2	N/A.	
*DLL1*	Delta-like ligand 1	Contributes to Notch signaling, which suppresses glioma stem cell differentiation and maintains their stem cell properties that contribute to GBM tumorigenesis.	Bazzoni et al., 2019 [20], Talukdar et al., 2016 [25]
*DLL3*	Delta-like ligand 3	An inhibitory ligand-driven activation of the Notch pathway and is a potent prognostic factor for malignant glioma; low DLL3 expression is linked to shorter overall survival.	Maimaiti et al., 2021 [26]
*DPP10*	Inactive dipeptidyl peptidase 10	Underexpressed in GBM but overexpressed in diffuse astrocytomas and anaplastic astrocytomas.	Gonzalez-Garcia et al., 2020 [51]
*ENHO*	Adropin (energy homeostasis-associated protein)	N/A.	
*ETNPPL*	Ethanolamine phosphate phospholyase	Underexpressed in GBM but overexpressed in diffuse astrocytomas and anaplastic astrocytomas.	Gonzalez-Garcia et al., 2020 [51]
*FERMT1*	Fermitin family member 1	N/A.	
*GDF10 (* *BMP3b)*	Growth differentiation factor 10	Associated with progression-free survival in GBM in a gender-dependent manner (PFS probability falls faster in males with high GDF10 expression than in females).	Serao et al., 2011 [47]
*IFGN1*	Immunoglobulin-like and fibronectin type III domain-containing protein 1	N/A.	
*IRX2*	Iroquois-class homeodomain protein IRX-2	Biomarker of pilocytic astrocytoma localization.	Antonelli et al., 2018 [52]
*LINC00836*	Long intergenic nonprotein-coding RNA 836	N/A.	
*LUZP2*	Leucine zipper protein 2	Crucial for nervous system extracellular matrix development; downregulated expression corresponds with increasing tumor stage in low-grade gliomas.	Chen et al., 2020 [48]
*MGAT4C*	α-1,3-mannosyl-glycoprotein 4-β-N-acetylglucosaminyltransferase C	N/A.	
*P2RY12*	P2Y purinoceptor 12	A specific marker for resident microglia in gliomas; its expression and localization correspond with tumor stage and M1/M2 immune responses.	Zhu et al., 2017 [53]
*SCG3*	Secretogranin III	Expression is inversely correlated with malignancy grade; high in oligodendrogliomas and low in GBM.	Wang et al., 2021 [27]
*SHANK2*	SH3 and multiple ankyrin repeat domains 2	N/A.	
*TNR*	Tenascin-R	Low expression in GBM; TNR dysregulation in GBM is associated with glioma malignancy.	Bi et al., 2017 [49]
*VIPR2*	Vasoactive intestinal polypeptide receptor 2	Overexpressed in gliomas, particularly in oligodendrogliomas.	Jaworski et al., 2000 [54]

**Table 5 ijms-23-04157-t005:** Clinical attributes of the GSE108474 metadata from the NCBI GEO repository.

Tumor Type	Number of Patients
Astrocytoma	148
Glioblastoma multiforme	228
Mixed	11
Oligodendroglioma	67
Unclassified ^1^	1
Unknown ^2^	67
Control ^3^	28

^1^ The type of glioma was unclassified for this patient. ^2^ The type of brain tumor was unknown in these patient samples. ^3^ Normal, healthy brain samples.

## Data Availability

The authors confirm that the data supporting the findings of this study are available within the article and its Appendix A.

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
