# Peer review of "Recognition of a Novel Gene Signature for Human Glioblastoma"

_ijms, 2022, doi:10.3390/ijms23084157_

Round 1

Reviewer 1 Report

The manuscript Recognition of a Novel Gene Signature for Human Glioblastoma provide a new point of reference for the prognostic prediction 
of glioblastoma, but also contribute to the understanding of molecular mechanisms in glioblastoma development.

I suggest to add in the section Introduction, a few clinical and histological images related to the glioblastoma. 

Regarding the Table 3 and 4, I suggest to mention in a separate columns the references and the first author.

Regarding the Table 5, I suggest to authors to explain a little bit the terms: Unclassified, Unknown, Control”.

Author Response

Responses to Reviewer Comments:

Reviewer #1

The manuscript Recognition of a Novel Gene Signature for Human Glioblastoma provide a new point of reference for the prognostic prediction of glioblastoma, but also contribute to the understanding of molecular mechanisms in glioblastoma development.

I suggest to add in the section Introduction, a few clinical and histological images related to the glioblastoma. 

Response:

We thank the Reviewer for these constructive comments. We have accordingly included some explanatory text in the Introduction section (in red font, lines 60-64) describing the benefits of clinical and histological images relating to the prediction of GBM prognosis.

Regarding the Table 3 and 4, I suggest to mention in a separate columns the references and the first author.

Response:

We thank the Reviewer for this thoughtful feedback. We have duly added a column denoted “References” to Tables 3 and 4, listing referenced work with surnames of the first authors.

Regarding the Table 5, I suggest to authors to explain a little bit the terms: Unclassified, Unknown, Control”.

Response:

We thank the Reviewer for this helpful advice. Notes have now been added to Table 5, explaining the terms “Unclassified”, “Unknown” and “Control”.

Reviewer 2 Report

The authors focus their study on glioblastoma by using an existing database in order to perform the GSEA and machine learning analysis in order to identify a 33 gene signature of the glioblastoma by examining astrocytoma or non GBM glioma differential gene expression.

 The manuscript is overall quality and easy to follow and the authors have well thought out their main contributions. The provided theoretical analysis is concrete, complete, and correct and the authors have provided all the details in order to enable the average reader to easily follow it.

The authors should consider the following suggestions provided by the reviewer in order to improve the scientific depth of their manuscript, as well as they need to address the following comments in order to improve the quality of the presentation of their manuscript.

Initially, in Section 1, the provided related work should be presented by using more summative language in order to better identify the research contributions that have already been performed in the literature and clarify what is the main research gap that the authors try to address.

In Section 1, the authors need to discuss how available data sets, such as Thai, My T., Weili Wu, and Hui Xiong, eds. Big data in complex and social networks. CRC Press, 2016, can support the proposed research or can even contribute in more complex solutions that can provide more accurate results by exploiting the big data.

Finally, the overall manuscript should be checked for typos, syntax, and grammar errors in order to improve the quality of its presentation

Author Response

Responses to Reviewer Comments:

Reviewer #2

The authors focus their study on glioblastoma by using an existing database in order to perform the GSEA and machine learning analysis in order to identify a 33 gene signature of the glioblastoma by examining astrocytoma or non GBM glioma differential gene expression.

The manuscript is overall quality and easy to follow and the authors have well thought out their main contributions. The provided theoretical analysis is concrete, complete, and correct and the authors have provided all the details in order to enable the average reader to easily follow it.

The authors should consider the following suggestions provided by the reviewer in order to improve the scientific depth of their manuscript, as well as they need to address the following comments in order to improve the quality of the presentation of their manuscript.

Initially, in Section 1, the provided related work should be presented by using more summative language in order to better identify the research contributions that have already been performed in the literature and clarify what is the main research gap that the authors try to address.

Response:

We thank the Reviewer for these constructive comments and suggestions. We have accordingly added text to paragraph 3 of the Introduction section, describing recently identified biomarkers and signaling pathways with reference to the literature (lines 68-72), to improve the scientific depth of our manuscript.

The main goal of our study is to use the machine learning technique to identify novel signature genes of GBM that not only enhance prognostic predictions for this disease, but also contribute to the understanding of molecular mechanisms in its development. Importantly, these novel signature genes may be exploited as therapeutic targets for GBM. No curative treatment strategies exist as yet for GBM. This suggests that identifying therapeutic strategies involving biomarkers with potential to destroy tumor cells in GBM remains an urgent task. We hope that the Reviewer agrees that the aims and significance of our study are clearly presented at the end of the third paragraph and also in the last paragraph of the Introduction section.

In Section 1, the authors need to discuss how available data sets, such as Thai, My T., Weili Wu, and Hui Xiong, eds. Big data in complex and social networks. CRC Press, 2016, can support the proposed research or can even contribute in more complex solutions that can provide more accurate results by exploiting the big data.

Response:

We thank the Reviewer for this suggestion. We have duly added a citation (Thai et al., 2016) illustrating how the processing of information from large-scale datasets helps us to interpret intelligence about life itself (lines 80-88).

Finally, the overall manuscript should be checked for typos, syntax, and grammar errors in order to improve the quality of its presentation

Response:

We thank the Reviewer for this reminder. The manuscript has been professionally edited by Iona MacDonald, who has amassed 25 years of experience in medical writing and editing of articles for many different scientific journals and also medical books. We believe that the manuscript is free of grammatical errors.

Round 2

Reviewer 2 Report

The authors have addressed the reviewers comments.